# LED Lighting to Produce High-Quality Ornamental Plants

**DOI:** 10.3390/plants12081667

**Published:** 2023-04-16

**Authors:** Alice Trivellini, Stefania Toscano, Daniela Romano, Antonio Ferrante

**Affiliations:** 1Department of Agriculture, Food and Environment, Università degli Studi di Catania, 95131 Catania, Italy; alice.trivellini@gmail.com; 2Department of Science Veterinary, Università degli Studi di Messina, 98168 Messina, Italy; stefania.toscano@unime.it; 3Department of Agricultural and Environmental Sciences—Production, Landscape, Agroenergy, Università degli Studi di Milano, 20133 Milan, Italy; antonio.ferrante@unimi.it

**Keywords:** light-emitting diodes, flowering, postharvest, plant architecture, red light, blue light, green light, ornamental plant production

## Abstract

The flexibility of LED technology, in terms of energy efficiency, robustness, compactness, long lifetime, and low heat emission, as well as its applications as a sole source or supplemental lighting system, offers interesting potential, giving the ornamental industry an edge over traditional production practices. Light is a fundamental environmental factor that provides energy for plants through photosynthesis, but it also acts as a signal and coordinates multifaceted plant-growth and development processes. With manipulations of light quality affecting specific plant traits such as flowering, plant architecture, and pigmentation, the focus has been placed on the ability to precisely manage the light growing environment, proving to be an effective tool to produce tailored plants according to market request. Applying lighting technology grants growers several productive advantages, such as planned production (early flowering, continuous production, and predictable yield), improved plant habitus (rooting and height), regulated leaf and flower color, and overall improved quality attributes of commodities. Potential LED benefits to the floriculture industry are not limited to the aesthetic and economic value of the product obtained; LED technology also represents a solid, sustainable option for reducing agrochemical (plant-growth regulators and pesticides) and energy inputs (power energy).

## 1. Introduction

The ornamental industry produces plants for their aesthetic value from a very wide range of species, including floriculture crops, ornamental grasses, turf grasses, and ornamental trees and shrubs. The economic impact of floriculture has remarkable importance. The worldwide flower and ornamental plant market amounted to as much as USD 52,384.85 million in 2022 and is expected to grow with a CAGR of 7.28% from 2023 to 2028 in the forecast period [1]. Europe is the largest ornamental market, with about 47% market share, followed by Asia–Pacific (20%) and the United States (13.5%) [1]. Despite the economic magnitude of the ornamental market, as with many nonessential commodities and industries, the COVID-19 pandemic and subsequent public shutdown events substantially damaged its production chain across the globe [2]. In addition, the ornamental sector deals with multiple challenges, including market globalization, uncertainty related to climate change, land-use competition, and anthropogenic pressures [3,4]. In this scenario, it is necessary to find more sustainable solutions for agricultural production sectors that allow for increases in the quality and yield of products while reducing production costs, environmental pressures, and natural resource depletion. For the optimization of ornamental production, in terms of both economics and sustainability, one opportunity/possibility is the application of artificial light in controlled environment systems (greenhouses, soilless systems, and indoor farming), which, overall, allows an accurate handling of environmental parameters by using a technology-driven approach [5].

Light is the essential environmental factor coordinating plant growth, development, and function since it represents the driving force for photosynthetic CO_2_ assimilation. It is also the signal that triggers multiple response pathways that are involved in many developmental aspects of growth, collectively recognized as photomorphogenesis [6]. In the last few decades, the use of artificial lighting for plant cultivation has become an interesting choice, either as a supplementary source when solar radiation is scarce, or as a sole light source, providing energy for photosynthesis, modulating crop morphogenesis, and regulating the flowering process [7,8,9]. By adjusting/modulating the light components/properties, such as quantity (intensity), duration (photoperiod), and quality (spectral composition), it is possible to attain important ornamental production targets to induce flowering, control leaf shape and plant architecture, extend the production season, fine-tune leaf and flower color, improve longevity, and enhance resilience to pathogens [6].

Until recently, the viable and widely used options for artificial lighting systems were high-intensity discharge (high-pressure sodium, HPS; metal halide, MZ) and fluorescent lamps due to their relatively high fluence and economical affordability [10]. However, these conventional lighting systems show some disadvantages, including the generation of excessive heat, high energy needs, and inability to modulate the light spectrum, generally emitting light over a limited broad spectrum (orange-red region, 550–650 nm, with less in the blue region, 400–500 nm) [11,12]. The recently emerged light-emitting diode (LED) technology has great potential for protected ornamental production [13]. LEDs offer several unique advantages over traditional lighting systems since they are the most energy-efficient and environmentally friendly lighting technologies currently available [14]. LEDs provide higher energy efficiency, which allows for reductions in electricity costs; together with their performance characteristics/features such as robustness, compactness, durability, and long lifetime, LEDs represent a cost-effective option that is largely appreciated in commercial settings [13,15]. Low heat emission allows the light source to be placed close to the canopy, ensuring a uniform spectral distribution while preventing tissue damage from photostress [16]. Additionally, owing to the advantages of high-light-intensity selection and spectral modulation, LEDs, remarkably, meet the specific requirements of leaf optical properties, encompassing dynamic photosynthetic activity and biochemistry processes to control the growth and development of plants [17].

This review provides an overview of the use of LED lighting technology for growing/producing ornamental crops (Figure 1).

The novelty of this review is to elucidate how LED illumination can be exploited to promote/prompt innovation in the ornamental market, while providing guidelines/recommendations for growers to improve the quality and yield of their production systems/practices. Therefore, the modulation of ornamental plant attributes by LEDs is discussed, including flowering regulation, plant architecture, postharvest/postproduction longevity, flower and leaf color, and pathogen and disease control (Table 1).

## 2. Flowering Regulation

Flower induction and initiation are complex processes driven by environmental and intrinsic factors that influence the transition from the vegetative phase to reproductive competence [43,44]. The integration of endogenous signals in response to external cues is strictly mediated by a complex network of genetic pathways to ensure the progeny’s success [45,46,47,48,49].

Various species of plants, including many ornamental crops, synchronize their growth and development by sensing changes in the light environment, such as photoperiod, light intensity, spectral composition, and direction [45]. In terms of photoperiodic requirements, most ornamental plants can be classified as long-day (LD) plants, short-day (SD) plants, and day-neutral plants (ND). Flowering of LD plants is induced when the night length is less than a certain threshold (critical duration). Flowering of SD plants is promoted during long nights (short days), whereas, in ND plants, flowering can occur irrespective of the day’s length [50]. During flowering, coordinated endogenous responses to the relative lengths of the light and the dark periods take place in leaves through a complex gene regulatory network involved in light sensing, which is driven by photoreceptor action [51,52]. Molecular evidence has demonstrated that the flowering transition occurs via upregulation of *FLOWERING LOCUS T (FT)*, also known as florigen, and repression of antiflorigenic *FT (AFT)/TERMINAL FLOWER 1 (TFL1)* [53]. The molecular mechanism of the inductive photoperiod is conserved in both LD and SD plants [54]. In addition to the photoperiod, the spectral composition, hormone pathways, and temperature play significant roles in the control of flowering for both LD and SD crops. In some ornamental crops, low-temperature exposure is required to regulate the transition from vegetative to reproductive growth (vernalization) [55]. In many species, the relationship between the photoperiod and the temperature has been shown to regulate the flowering transition, thus representing the main integrated approach for harvesting schedules and utilizing greenhouse space. This allows the planning of a controlled growing environment, as well as producing continuously predictable yields over predetermined time periods. Typically, a low light intensity is provided at night to boost the flowering of LD plants and reduce their crop production cycle, while preventing flowering in SD plants and promoting their vegetative growth [54]. Chrysanthemums are the second most important ornamental crop and are grown as cut or potted flowers. To ensure year-round availability for the market demand for short-day plants such as chrysanthemums, as well as to assure the programmed flowering on predetermined market dates, artificial lighting is provided as a day-length extension to promote vegetative growth or as a night break to prevent premature flowering [56,57].

In addition to the photoperiod, the spectral composition influences the flowering process in short-day and long-day plants [58]. The specific light quality drives the flower transition, which in turn leads to transcriptional regulation of the genes that encode activators of flowering, i.e., the photoreceptors. Several photoreceptors are involved in the perception and absorption of different wavelengths: phytochromes that preferentially absorb in the red (660 nm)/far-red (730 nm) spectral regions; cryptochromes that preferentially absorb in the blue/UV-A wavelengths; and phototropins (PHOT), ZTL/FKF1/LKP2, and UVR8 that mostly absorb UV-B light [59]. Blue light and far-red light are typically effective in promoting flowering in LD plants (Figure 1 and Figure 2). The efficacy of LEDs compared to conventional lamps (HPS) was evaluated by comparing their regulatory role in the flowering of photoperiodic plants. In this respect, LEDs provide comparable effectiveness to conventional light sources, while featuring a lower total operating cost [60]. In petunias (*Petunia hybrida* E. Vilm.) and snapdragons (*Antirrhinum majus* L.), long hours of illumination using a high daily light integral (DLI) and a red/white/far-red lamp significantly encouraged flower formation and development [61]. Similarly, in day-neutral *Cyclamen persicum* Mill., the combined use of high light intensity with blue and red wavelengths was useful to promote flowering and subsequent development [25]. On the other hand, when LD crops (i.e., snapdragon, tussock bellflower, tickseed, and petunia) were grown under a far-red light-deficient environment, a delay in flower initiation and development was observed [62,63,64]. SD plants cultivated in greenhouses are negatively affected by the lower DLI and shorter photoperiod occurring over the winter season. However, supplemental illumination that prolongs the day length by using far-red light has been reported as a cost-effective strategy that favors growth extension, e.g., as a tool to improve the plant habitus.

When using end-of-day far-red treatments in poinsettias [65], chrysanthemums, and garden strawberries [66], flowering initiation occurred later in development, while the plants showed longer stems and longer internodes. The delayed flowering status may be attributed to the altered phytochrome level in a far-red light environment at the end of the day [66]. A similar inhibitory effect was reported in chrysanthemums, where short days of solar light, followed by a 4 h extension with blue or red light, were not enough to affect the flowering initiation [67]. When the natural day is short, the use of red light to interrupt the night is a typical practice to inhibit flowering in SD plants due to the photochemical interconversion of phytochrome Pr to the Pfr form during the night [68]. Moreover, by applying far-red light, the flowering inhibition mediated by the phytochrome photoequilibrium was reversed [69]. Furthermore, blue light, as well as its light signaling initiated by the cryptochrome, has a significant impact on flowering and can be used to control the process. At a higher intensity (20 µmol∙m^−2^∙s^−1^ or higher), blue light has a flower-promoting effect on LD plants, while showing an inhibitory action on SD plants when applied as a night-break or day-length extension regime [70,71]. The promotion effect of blue light on LD flowering was observed as earlier flowering, a greater flowering index, and more visible flower buds and opened flowers, seemingly associated with lower phytochrome activity, also known as a phytochrome photostationary state [35].

The effect of green light on photoperiodic flowering has been reported in a few studies. Similar to blue-light flowering responses, short-day plants grown in a green light environment showed a delay/inhibition of flowering, depending on the species, as well as the duration and intensity of exposure [21,70,72] (Figure 3). Moreover, Meng and Runkle [32] showed that fluxes of green radiation may function as a long-day signal. Adopting a spectrum with moderate intensities of green light for several hours was effective in saturating the flowering responses of petunias, snapdragons, and ageratum floriculture crops. On the other hand, in chrysanthemum and marigold SD plants, the delivery of green light exerted a delayed flowering effect, suggesting a role for this wavelength in the control of flower induction for photoperiodic plants [32].

Application of a UV spectrum can either promote or delay flowering; the responses depend on the species, region of the UV spectrum, and fluence rate. With regard to this last aspect, high UV radiation has been shown to dramatically impact flowering quality (Figure 4). For example, the flowering time and the number of flowers produced in *Phacelia campanularia* A. Gray and *Salvia splendens* Sellow ex Nees plants exposed to high UV dosages were significantly hampered [73,74], whereas the flower transitions of *Limnanthes alba* Hartw. ex Benth. plants were inhibited [73]. In contrast, UV-C radiation improved flowering and even increased the flower number in wild pansy and freesia ornamental plants [74,75].

## 3. Plant Architecture

LED technology, through the ability to select specific wavelengths, offers the possibility to develop tailored light recipes for the manipulation of plant architecture. Plant quality (distribution of energy across different wavelengths) is often a mix of specific plant traits, such as branching, compactness, rooting, and leaf expansion, which are strongly influenced by the spectral composition of LED light [11]. The reasonable choice for commercial plant production using LED systems is the combination of red and blue wavelengths, since the absorption spectra of photosynthetic pigments mainly focus on blue (400–500 nm) and red (600–700 nm) light [76], and several regulatory mechanisms can be exploited.

Young ornamental plant production is an integral part of the floriculture industry. The bulk of production occurs in winter or early spring to meet the spring and summer sale demand. Unfortunately, this is also when the outdoor photosynthetic daily light integral (DLI) is seasonally low, and is even lower in greenhouses. One of the most cost-effective applications of LED lighting is bedding plant production, which allows the obtainment of more uniform, compact, and high-quality annual young ornamental plants with marketable characteristics, as well as the ability to withstand transplanting shock. Controlling the growth of these commodities is a vital aspect of the ornamental industry since this allows the improvement of both their visual quality and their physiological status. Several studies have used red and blue light to assess their effects on the morphology and anatomy of plants. In general, red and blue LED lights affect physiological and morphological traits, such as stomatal openings, plant height, chlorophyll biosynthesis, stem elongation, branching, leaf expansion, and reproduction [11]. Both supplemental and sole sources of LED lighting, with blue radiation in a red background, limit the extension growth and leaf expansion compared to growth under ambient light supplemented with an HPS lamp or cool white fluorescence, providing an effective nonchemical method to control the height of several species of bedding plants [12,30,77]. Furthermore, the productivity and quality of cuttings can be modulated/influenced by LED treatment. Adventitious rooting is a critical process in the vegetative propagation of ornamental plants, and LED lighting positively affects the growth, survival, and rooting of cuttings. Compared to the application of red or blue light alone, the combined use of red and blue light (R:B ratio of 1:1) in a multilayer sole-source light propagation system reduced stem elongation and improved root biomass in herbaceous perennial cuttings, while avoiding damage during shipping and transplanting [40]. Cuttings are susceptible to fast drying, and the control of transpiration can be achieved using a well-balanced light spectrum, since the stomatal opening response is predominantly initiated by blue light but enhanced under a strong red-light background [78,79]. In *Impatiens ×hybrida* hort., for instance, environments with a high percentage of red light but a lower percentage of blue light have been shown to increase the number of trichomes, anatomical structures linked to the prevention of water loss by transpiration [29]. Additionally, this light recipe led to a greater plug compactness and survival of cuttings, achieving a tradeoff between the risk of dehydration and quality of cuttings. In other species, such as *Chrysanthemum ×morifolium* (Ramat.) Hemsl., *Lavandula angustifolia* Mill., and *Rhododendron simsii* Planch. hybrids, treatment with red light only (100) was highly efficient in enhancing rooting performance [20].

The cultivation of cut flowers in a protected environment enables year-round production in northern latitudes, which are characterized by unfavorable conditions. In fact, using LED lighting solutions as a supplemental source in greenhouses overcomes the concrete risk of not reaching the minimum lighting requirements for crops [80]. The shoot architecture, particularly stem elongation, can be regulated by controlling the shade avoidance phenomenon related to the excessive growth of plants when subjected to the shade of other plants or when growing in high-density conditions, whereby the availability of photosynthetically active radiation (PAR) is reduced, along with the ratio of red-to-far-red (R/FR) light [81]. Along with physiological changes, a low phytochrome stationary state enhances internode and petiole elongation, axillary bud outgrowth, and hyponasty [81]. Grading standards for cut chrysanthemum flowers on the world market require an elongated and unbranched plant shape and large-sized flowers. Treatment of rooted chrysanthemum cuttings with a combination of blue and far-red light showed a higher internode length compared to sole red light; in decapitated cuttings, the apical bud concomitantly reached a high length with inhibited growth of underlying buds [82]. In lilium, grown as a cut flower, different ratios of red to blue light have been shown to influence different characteristics; when exposed to the highest red percentage (R:B ratio of 80:20), the height of the stems was greatly enhanced [33]. Furthermore, upon increasing the percentage of blue light, several morphological traits were modulated, such as reduced time to harvest (R:B ratio of 20:80), strong inhibition of stem elongation (R:B ratio of 40:60), and slightly improved vase life (R:B ratio of 60:40) [33]. Similarly, potted miniature rose ‘Aga’ plants, grown under a supplemental wide spectrum of red, blue, white, and far-red LEDs, exhibited significantly greater height and shoot length than control plants [37]. Exposure to blue and blue/red light positively affected the photosynthetic performance of *Cordyline australis* (G. Forst.) Endl., *Ficus benjamina* L., and *Sinningia speciosa* Hiern potted foliage plants, while also showing greater stomatal conductance and density, as well as an increase in leaf thickness [23]. Generally, the commercial standard for potted plants requires a compact shape; thus, the light spectrum tends to be manipulated while avoiding light sources with a low R:FR ratio since its effects on overall growth reduces the decorative value of ornamental potted plants. In petunias, despite the promoting effect of a lower ratio of red-to-far-red light on flowering, the overall quality was adversely affected, showing excessive stem elongation, weak stems, and poor branching [83]. However, by adding a moderate green wavelength to a red, far-red, and white background, petunia plants were shorter and developed more branches [60]. Green light has been shown to participate in regulating growth and development through its ability to much more effectively penetrate the lower canopy, thereby optimizing the photosynthetic machinery [84]. The use of pure green light in a recent study promoted elongation via brassinosteroids, triggering the activation of the *BRI1-EMS-SUPPRESSOR 1 (BES1*) transcription factor and the target genes in its downstream signaling pathway [19]. This effect of green light on stem elongation has been reported, for example, in *Zingiber officinale* Roscoe, where even the use of supplemental, green-enriched light enabled the plants to strongly improve their photosynthetic performance [85].

## 4. Postharvest/Postproduction Longevity

The quality of ornamental plants depends not only on their external attributes, such as shape, size, color, and flower and leaf turnover, but also on the ability to preserve their characteristics [3]. In fact, as a fresh commodity, they are still metabolically active and extremely perishable after harvest/production and are highly vulnerable to large postharvest losses. Thus, the longevity of ornamental plants is the main goal for their commercial success, but suboptimal postharvest conditions that often occur during storage and transportation negatively impact the overall quality and accelerate degenerative processes. Regardless of the product type (i.e., cut flowers, potted foliage, and flowering plants), the main postharvest disorders that compromise the decorative value are leaf yellowing, flower and bud senescence, and abscission [3,86]. To avoid a negative impact on the marketability of these commodities and the resulting reduction in profit for both producers and sellers, several postharvest handling approaches have been developed over the years. Technological interventions have mainly focused on the structural optimization of the postharvest chain (transport and storage), as well as the development of novel packaging and precondition techniques to delay the senescence process [86]. Several commercial chemicals are available for cut flowers and potted plants, including ethylene inhibitors, antibacterial agents, synthetic growth regulators, and sugars. Ethylene is widely known as an aging hormone, as it promotes and accelerates senescence-related processes. To prevent leaf and flower abscission, the application of gaseous 1-methylcyclopropene (1-MCP) was effective in blocking ethylene perception and prolonging the longevity (maintaining the freshness) of many ethylene-sensitive potted plants and cut flowers [87]. As biosynthesis inhibitors, amino vinylglycine (AVG) and amino-oxyacetic acid (AOA) avoid endogenous ethylene production by interfering with the key ethylene biosynthetic enzyme 1-aminocyclopropane-1-carboxylic acid (ACC) synthase (ACS, E.C. 4.4.1.14), without impeding the deterioration process trigged by exogenous ethylene [87]. Furthermore, the synthetic compound thidiazuron, with a strong cytokinin-like activity, was successfully used in cut foliage and flowers, as well as in potted plants, to improve postharvest performance (i.e., delaying leaf yellowing and increasing flower longevity) [3,87]. In addition to the crucial role of chemical preservatives, control of the storage and transportation environment assumes great relevance within the ornamental industry [88]. In fact, higher temperatures during storage and transportation are linked to enhanced respiration rates and ethylene production, physiological processes that highly accelerate plant senescence, thereby negatively influencing the overall quality of ornamental products. For this reason, the logistic chain process from the producers to the storage and transportation facilities, and then to the end-user/consumer, is committed to maintaining a cold temperature. However, among the various handling processes of postharvest chains, ornamental products are exposed to fluctuating temperatures, and keeping an optimally low temperature requires high equipment and operating costs. The development of novel systems focused on the postharvest storage environment by manipulating the light spectrum might represent an alternative strategy to support the quality of ornamental products [88]. Recently, the processes underlying the vase life response under different light environments were explored in several ornamental plants. In anthuriums, a tropical cold-sensitive species, prolonged exposure to a low storage temperature compromised the longevity of its cut stems [18]. Applying a sole source of blue light under cold storage led to the highest rate of water loss and electrolyte leakage from the spathes, whereas red light lowered the ROS accumulation in the cells [18]. These metabolic adjustments also led to a significant reduction in vase life in cold storage under blue light, suggesting that oxidative stress and membrane integrity negatively affected the cold tolerance and quality of anthurium spathe. In contrast, cut carnation flowers exposed to blue light showed a prolonged vase life and a markedly delayed senescence [26]. Simultaneously, increases in photosynthetic performance, transpiration rate, sugar content, and water uptake were recorded. These results were in agreement with previous studies on the positive role of blue light in stomatal movements, suggesting a better water transport efficiency and a preserved photosynthetic ability of leaves [89,90,91]. Moreover, the higher antioxidant capacity reported could guarantee a higher membrane integrity and ROS detoxification, consequently preserving/maintaining photosynthetic ability [26] and confirming that the evaluation of antioxidant status in ornamental products represents a key qualitative index to monitor postharvest [26]. The effect of blue light was also investigated in relation to the expression pattern of genes involved in ABA homeostasis, ethylene biosynthesis, and signaling in carnation cut flowers [26]. Both hormones are well known to play regulatory roles during flower senescence in ethylene-sensitive and nonsensitive species [92,93,94,95]. Blue light exposure during storage reduced the expression of the *ACS1* and *ACO1* genes involved in ethylene biosynthesis, whereas red light led to an increase in their levels [92]. Their relative transcript abundance correlated with their longevity; blue light-exposed cut flowers showed a superior/improved vase life to red light-exposed flowers. Moreover, the activation of transcriptional pathways related to ABA biosynthesis and its transport by blue light appeared to significantly enhance the vase life of cut carnation flowers. Red light has been shown to influence water balance and flower opening in cut roses [88]. In particular, the petal fresh weight was significantly higher under red light compared to the other treatments (blue and white light), and the cut flowers showed a longer vase life. Thus, exposure to a specific light wavelength, particularly red light, might be an effective tool to control flower opening and longevity. LED lighting was found to efficiently maintain plants in an indoor environment distinguished by a low light intensity. Supplemental LED lighting was positioned upward around the base, providing the whole range of radiation (300–800 nm), delaying the senescence of lower leaves, and encouraging flower opening in potted rose plants compared to the downward setting, suggesting a suitable arrangement for indoor plant management [39].

## 5. Flower and Leaf Color

Another important quality index for the ornamental plant industry, in addition to plant architecture and longevity, is the color of the leaves and flowers, which guides consumers toward their preferences and, therefore, their purchases. The major classes of plant pigments that determine foliage and flower color are chlorophylls, carotenoids, anthocyanins flavonoids, and betalains [96]. Environmental conditions (i.e., temperature, light intensity, and light spectrum) and genetic determinants that mainly drive the development and regulation of pigmentation patterns. Regarding light intensity, plants fall into three categories: high, medium, and low light requirements. The flowers of tuberose and boronia plants, when grown under full sunlight conditions, develop an intense reddish-purple pigmentation compared to those obtained under a shaded environment [97,98]. Similarly, faded flowers were observed when peony flowers were developed in a partially shaded environment due to the downregulation of phenylalanine ammonialyase (*PAL E.C. 4.3.1.24*) and chalcone synthase (*CHS* E.C. *2.3.1.74*) genes, which are key regulatory steps in the anthocyanin biosynthesis pathway [99]. Furthermore, light has been reported to modulate the accumulation of pigments in various plant organs. For example, in *Hibiscus syriacus* L. flowers, red light exposure influenced the development of a strong red color in their petals [100]. During winter, the aesthetic quality of some potted foliage plants is negatively affected by the low light intensity since the full biosynthesis of pigments is impeded. The use of LED supplemental lighting enriched with red and blue wavelengths has been reported to increase the accumulation of anthocyanins and carotenoids, leading to a vivid foliage color and an overall improvement in plant’s decorative value [28]. In geraniums and purple fountain grass plants, the supplementation of red and blue LED light at the end of production significantly enhanced red color saturation, expressed as the chroma index, thus increasing aesthetic appeal, quality, and market value [40].

## 6. Pathogens and Disease Control

Light not only regulates the primary/basic metabolic functions and development (flowering, stem elongation, and morphology) of plants, but can also play a key role in the regulatory network involved in secondary metabolite biosynthesis and accumulation through modulation of the photosensory signaling pathway, orchestrated by photoreceptors [101]. The functional and biological roles of secondary metabolites vary, including defense against phytophages, intra- and/or interspecies communication, protection against harmful solar radiation, and signals for pollination or seed dispersion [102]. Manipulating the light spectrum constitutes an interesting elicitation strategy that actively and suitably interferes with biosynthetic routes, enhancing the concentration of key phytochemicals that can be exploited to improve the growth performance and the final quality of ornamental products (e.g., increasing plant fitness) [11,24,41]. Moreover, light quality, in addition to causing differential metabolic rearrangements, can directly or indirectly impact pathogens and pests, as well as their natural antagonists. Limited exposure to a UV-B light fluorescent lamp is often used to reduce disease incidence in crops that are cultivated in controlled environments (such as growth chambers and greenhouses). In high-density greenhouse-grown roses, low doses of UV-B radiation applied for 6 h completely reduced powdery mildew infection by increasing secondary bioactive compounds [103]. The use of red LED light was instead reported to reduce the number of conidia, suggesting its interesting potential for controlling powdery mildew disease in roses [38]. Gray mold caused by the fungus *Botrytis cinerea* is one of the most common and destructive plant pathogens, affecting several horticultural crops. Blue light and UV radiation have been shown to suppress the pathogenic development of *B. cinerea* on harvested vegetables, highlighting their potential application for a wide range of gray-mold-sensitive ornamental crops [88,104]. To limit the damage to flowering plants caused by the behavior of nocturnal moths, LED lights were used as a cheaper pull-and-push strategy against the insects compared to incandescent lamps [105]. Recently, the biological control of thrips and pests has gained popularity as a novel greenhouse system. In Europe, the arthropod *Orius laevigatus* is available on the market for thrips control in chrysanthemum crop production. *O. laevigatus* has a relatively quick population buildup, and thus is suitable for the short cropping cycle of chrysanthemums, as long as unfavorable environmental conditions are monitored [106]. In chrysanthemum production, after the introduction of a biocontrol agent for efficient population density establishment, different spectral wavelengths were evaluated in terms of egg-laying activity. The use of red, blue, and green light in equal proportions positively affected the number of eggs laid by *O. laevigatus,* whereas red light led to the lowest number of eggs [106].

## 7. Conclusions

The careful selection of components of the light spectrum by using LED lighting technology can significantly improve the quality-related properties/characteristics of ornamental products by influencing several physiological and metabolic processes, such as flowering, branching, rooting, pigment biosynthesis, and vase life. The effects of this technology can vary depending on ornamental species, exposure time, and applied wavelengths; thus, the identification of specific/optimal light formulas is fundamental to achieve the best results. The manipulation of flowering can help reduce costs and production time while obtaining a predictable yield, shaping/modeling the plant habitus and emphasizing attractive features. Moreover, artificial lighting offers a potential alternative to growth retardants used in chemical pinching, as well as an interesting tool for the control of some plant pathogens in greenhouses or growth chambers. The use of LED light in controlled environments can lead to the production of ornamental products with superior characteristics, representing a new frontier of applied sciences with studies focused on species/cultivar-specific light requirements. Furthermore, its application can help in reducing the use of agricultural inputs such as energy and soil in a sustainable manner.

## Figures and Tables

**Figure 1 plants-12-01667-f001:**
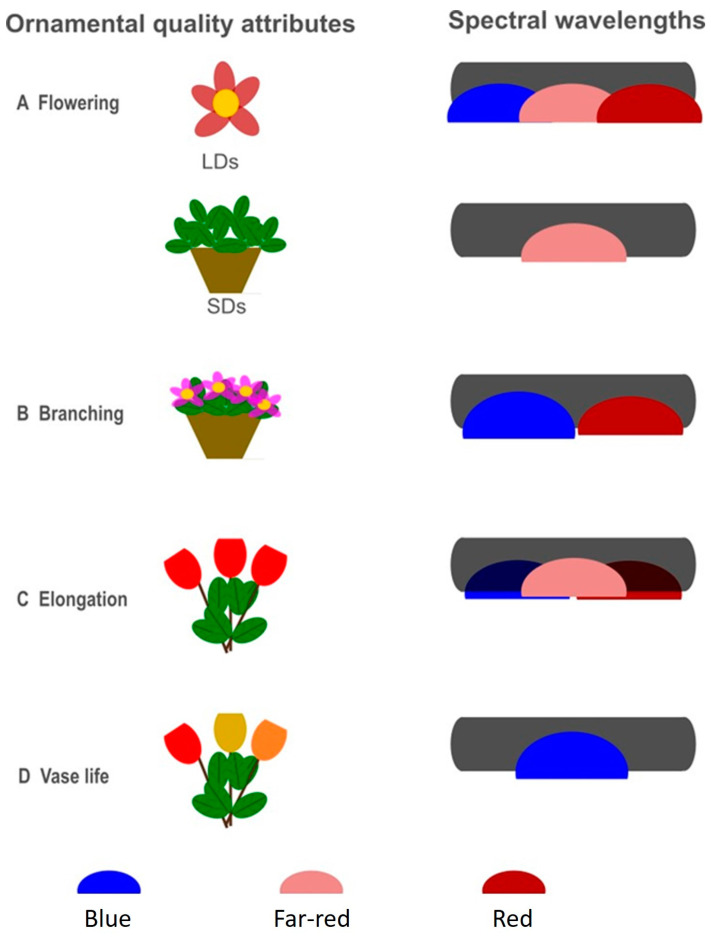
Spectral wavelengths affecting decorative traits of ornamental plants. (**A**) Blue, far-red, and red lights are typically effective in promoting flowering in long-day (LD) plants, whereas growth extension in short-day (SD) plants is promoted by using supplemental illumination at the end of the day with far-red light. (**B**) Blue radiation in a red background limits extension growth and promotes branching. (**C**) Stem elongation is regulated by controlling the shade avoidance phenomenon, using a lower R/Fr ratio or higher red percentage in a blue-light environment. (**D**) The vase life of cut flowers is improved when stored in a cold room and exposed to a sole source of blue light.

**Figure 2 plants-12-01667-f002:**
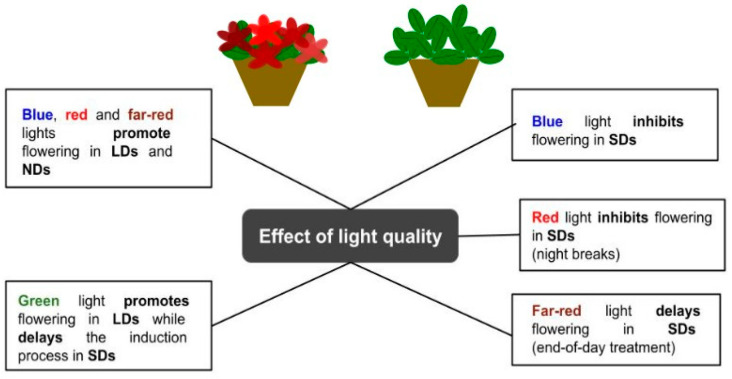
Schematic representation of the effect of light quality on the flowering process of ornamental plants.

**Figure 3 plants-12-01667-f003:**
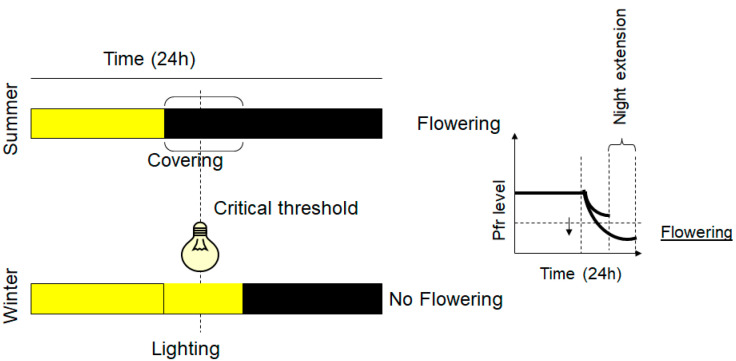
SD plants and flowering control. Yellow color represents daylight or lighting and black color the night or dark induced by covering the plants. LED light with emission at 730 nm converts the Pfr into Pr and inhibits the flowering of SD plants, overcoming the critical time threshold. On a biochemical level, right graph, nighttime reduces the Pfr level, and the induction of SD plant flowering occurs when the Pfr level declines below a critical concentration threshold. Night extension by covering the plants induces flowering and it is a strategy used in summer for inducing flowering of SD plants.

**Figure 4 plants-12-01667-f004:**
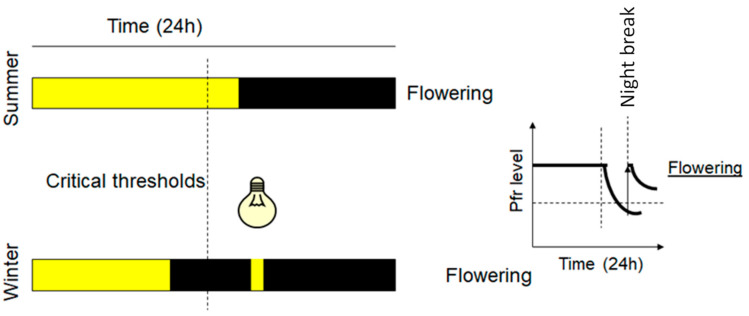
LD plants and flowering control. Yellow color represents daylight or lighting and black color the night or dark induced by covering the plants LED light with emission at 730 nm converts the Pfr into Pr and induces the flowering of LD plants, overcoming the critical time threshold. On a biochemical level, right graph, short (flash) lighting induces night break and increases the Pfr level, and the induction of LD plant flowering occurs when the Pfr level exceeds a critical concentration threshold.

**Table 1 plants-12-01667-t001:** Main effects of LEDs on different ornamental plants.

Species	Light Typologies	Effect of LEDs on Plants	References
*Anthurium andraeanum* Linden ‘Calore’, ‘Angel’	Darkness (D); different light spectra (R, B, RB (70:30%), and W) at 125 µmol.m^−2^ s^−1^.	B and W increased electrolyte leakage (EL); R decreased EL; B increased water loss; D and R decreased water loss. Negative correlation for both cultivars between EL and vase life and anthocyanin concentration and EL, and a positive correlation between anthocyanin concentration and vase life, were found. Higher percentage of B spectra determined higher EL and a shorter vase life under a cold storage condition.	[18]
*Arabidopsis thaliana* (L.) Heynh.	Blue (B), red (R), far-red (F:R), UV-B light, and green (G) light sources.	G promoted hypocotyl elongation, and the brassinosteroid (BR) signaling pathway is involved in this process. G promoted the DNA binding activity of BRI1-EMS-SUPPRESSOR 1 (BES1), thus regulating gene transcription to promote hypocotyl elongation.	[19]
*Chrysanthemum* ×*morifolium* (Ramat.) Hemsl., *Lavandula angustifolia* Mill., and *Rhododendron simsii* Planch. hybrids	R:B 100:0, 90:10, 80:20, 50:50, 10:90 and 0:100 at a light intensity of 60 µmol m^−2^ s^−1^ for *Chrysanthemum* ×*morifolium* and *Lavandula angustifolia* and 30 µmol m^−2^ s^−1^ for *Rhododendron simsii* hybrids.	R 100% increased root formation. 10:90 R:B inhibited rooting in *Chrysanthemum* ×*morifolium*, while under 50:50 R:B was inhibited rooting in *Rhododendron simsii*.	[20]
*Chrysanthemum ×morifolium* (Ramat.) Hemsl. ‘Gaya yellow’	Plants were grown under supplemental B (463 nm), G (518 nm), R (632 nm), and W LEDs.	W increased the weights of leaves and stems. G increased polyphenols (luteolin-7-O-glucoside, luteolin-7-O-glucuronide, quercetagetin-trimethyl ether); R increased dicaffeoylquinic acid isomer, dicaffeoylquinic acid isomer, naringenin, and apigenin-7-O-glucuronide.	[21]
*Chrysanthemum morifolium* ‘Orlando’	Blue, red, far-red; daily light integral: 4.1 mol m^−2^ d^−1^ in interaction with auxin treatments.	Lowering the R:FR ratio improved rooting significantly. In contrast, adding blue light to solely red light decreased rooting. Phytochrome plays a role in adventitious root formation through the action of auxin, but the blue light receptors interact in this process.	[22]
*Cordyline australis* (G. Forst.) Endl., *Ficus benjamina* L., *Sinningia speciosa* Hiern	B (100% blue, 460 nm), R (100% red, 660 nm), and W (white, 7% blue (400–500 nm), 16% green (500–600 nm), 75% red (600–700 nm), and 2% far-red (700–800 nm)) and RB (75% R and 25% B, peaks at 460 and 660 nm).	B and RB increased Fv/Fm and ΦPSII; R decreased biomass. B increased stomatal conductance, leaf thickness, and palisade parenchyma in *F. benjamina*. B and RB increased palisade parenchyma in *S. speciosa*.	[23]
*Crocus sativus* L.	(i) R ʎ = 660 nm (62%) and B ʎ = 450 nm (38%) (RB); and (ii) R ʎ = 660 nm (50%), G ʎ = 500–600 nm (12%), and B ʎ = 450 nm (38%) (RGB) and a photosynthetic photon flux density of 120 µmol m^−2^ s^−1^.	The two LED treatments increased the antioxidant compounds. RGB enhanced the total flavonoid content and declined corolla fresh weight. RB and RGB increased DPPH.	[24]
*Cyclamen persicum* Mill. ‘Dixie White’	B light treatment; R light treatment; mixing of B and R (BR) light treatments (1:1 photon flux density). Photoperiod of 10 or 12 h per day.	BR improved flower induction, with number of flower buds and open flowers being highest in the plants grown under RB (10 h per day). B and R alone reduced the flowering response. Peduncle length and blooming period of flowers were also influenced by light qualities and photoperiod treatments. Red length increased peduncle length. R increased the blooming period.	[25]
*Dianthus caryophyllus* L. ‘Moon light’	W (400–730 nm), B (460 nm), and R (660 nm).	B maintained a higher membrane stability index; higher activities of SOD, POD, CAT, and APX; a decline in petal carotenoid; a higher Fv/Fm percentage of open stomata; and a higher sugar content.	[26]
*Dianthus caryophyllus* L.	W (400–730 nm), R (660 nm), and B (460 nm).	B determined the lowest relative membrane permeability (RMP) in flowers, and longest vase life. The R and W lights accelerated flower senescence and increased expression of DcACS and DcACO. B inhibited the expression of ethylene biosynthetic genes.	[27]
(i) *Hypoestes phyllostachya* Baker ‘Decor Pink’ and ‘Decor Red’, *Guzmania lingulata* Mez. ‘Theresa’; (ii) *Cryptanthus carnosus* Mez. ‘Tricolor’	(i) 100R0B, 80R20B, 50R50B, 20R80B, and 0R100B (ii) 100R, 100R + FR, and 83R17B and 86R14B.	R and B are needed to preserve plant quality. In *Hypoestes*, the R LEDs determined curly leaves and plants that were not sufficiently compact. Without B light in *Guzmania*, bracts turn entirely yellow and *Cryptanthus* leaves are much paler. The B light improves the anthocyanin synthesis and qualitative pigmentation.	[28]
*Impatiens hybrida* hort (‘Sunpatiens Compact Royal Magenta’ = Magenta and ‘Sunpatiens Compact White’ = White)	83%R:17%B; 75%R:25%B; 67%R:33%B; and 50%R:25%R:25%B.	75R:25B and 83R:17B increased the cutting number in both cultivars. White cv. produced a higher number of cuttings compared to magenta, but only at 83 DAT in the 67R:33B treatment. At 167 days, 83R:17B produced a higher number of cuttings than 67R:33B. At 202 days, 83R:17B improved the number of cuttings compared to control. 67R:33B and 83R:17B increased leaf trichome numbers compared to the control.	[29]
*Impatiens walleriana* Hook.f., *Salvia splendens* Sellow ex Nees, *Petunia hybrida* E. Vilm.	B100, B50 + G50, B50 + R50, B25 + G25 + R50, G50 + R50, and R100, with a photosynthetic photon flux of 160 µmol·m^−2^·s^−1^ for 18 h·d^−1^.	For all species, plants grown under 25% or greater B light were shorter than those under R light. For all species, the plants under R light increased leaf area and fresh shoot weight more than plants grown under treatments with 25% or greater B light. B increased in *Impatiens walleriana* the flower bud.	[30]
*Lachenalia* spp.	Three light treatments: red (660 nm) and blue (440 nm) lights in different ratios: 100% R (100/0), 90% R + 10% B (90/10), and 80% + 20% B (80/20). The PPFD at the plant leaf level was approx. 150 μmol m^−2^ s^−1^.	The 90/10 spectrum induced the longest inflorescences with the highest stem diameter and number of florets. B light increased the anthocyanin content in the corolla (+~35%) compared to plants exposed to 100% R light and nonirradiated ones (control plants).	[31]
LDPs (long-day plants): two petunia cultivars, ageratum, snapdragons, and *Arabidopsis*; and SDPs (short-day plants): three chrysanthemum cultivars and marigold	Greenhouse undertruncated 9 h short days with or without 7 h day-extension lighting from G (peak = 521 nm) at 0, 2, 13, or 25 μmol m^−2^ s^−1^ or R + W + FR light at 2 μmol m^−2^ s^−1^.	Increasing the G photon flux density from 0 to 25 μmol m^−2^ s^−1^ accelerated flowering of all LDPs and delayed flowering of all SDPs. Petunias flowered similarly fast under R + W + FR light and moderate G light; under G, petunia plants were shorter and developed more branches. To be as effective as the R + W + FR light, saturation of G photon flux densities were 2 μmol m^−2^ s^−1^ for ageratum and marigold and 13 μmol m^−2^ s^−1^ for petunias. Snapdragons were the least sensitive to G. In *Arabidopsis*, cryptochrome 2 mediated the promotion of flowering under moderate G, whereas both phytochrome B and cryptochrome 2 mediated that under R + W + FR light.	[32]
*Lilium* spp. ‘Corvara’	20:80 (R4B); 40:60 (2R3B); 60:40 (3R2B); 80:20 (4RB); and control (W) (100% white light).	2R3B reduced the number of days to harvest maturity and flower height. Control increases were achieved in the following variables: R4B = leaf area, tepal color; 3R2B = vase life; and 4RB = plant height, flower diameter, and number of days to maturity.	[33]
*Petunia hybrida* E. Vilm., Geranium (*Pelargonium ×hortorum* L.H. Bailey), and Coleus (*Solenostemon scutellariodes* (L.) Codd)	R:FR (1:0, 2:1, and 1:1) at two PPFDs (96 and 288 μmol m^−2^ s^−1^), all with a B photon flux density of 32 μmol m^−2^ s^−1^.	As R:FR decreased, stem length in all species increased. Decreasing R:FR increased the leaf area in petunias, and increased shoot dry weight in petunias and coleus. Decreasing R:FR promoted in petunias subsequent flowering at both PPFDs. In geraniums, the addition of FR had no effect on flowering, irrespective of PPFD.	[34]
*Petunia hybrida* E. Vilm. ’Duvet Red’, *Calibrachoa ×hybrida* ‘Kabloom Deep Blue’, *Pelargonium ×hortorum* L.H. Bailey ‘Pinto Premium Salmon’, and *Tagetes erecta* L. ‘Antigua Orange’	R (660 nm); B (455 nm); BRF0; BRF2; BRF4; and BRF6. Unpure B light was created by mixing B with low-level (6%) R, and further adding far-red light of 0, 2, 4, and 6 μmol m^−2^ s^−1^, respectively.	B and BRF6 promoted flowering compared to R or BRF0. The promotion effect of unpure B light increased, following the order of BRF0, BRF2, BRF4, and BRF6, which varied in sensitivity among plant species. The calculated phytochrome photostationary state was higher for R and decreased gradually for unpure blue light treatments: BRF0, BRF2, BRF4, and BRF6.	[35]
*Rosa* ‘Red Star’	R (660 nm), B (440 nm), W (white); and darkness (dark).	W increased water uptake and evaporation rates; water uptake and evaporation did not modify the quality of cut roses subjected to red light treatment.	[36]
*Rosa* ×*hybrida* ‘Aga’	R, B, W, RBW + FR (far-red) (high R:FR), and RBW + FR (low R:FR).	Both RBW + FR lights increased plant growth and total shoot length. Light treatments increased Fv/Fm. R and RBW + FR at high R:FR stimulated flower bud formation. R increased the resistant to *Podosphaera pannosa.* B increased the flavonol index.	[37]
*Rosa* ×*hybrida* ‘Mistral’	To study the effects of light quality and light intensity on conidial productivity, chambers for conidia (*Podosphaera pannosa*) production were equipped with LEDs of B (465 nm), R (675 nm), F-R (755 nm), or W (full spectrum) in confront of mercury lamps (white light source).	The number of conidia trapped under F-R LEDs was approximately 4.7 times higher than in W light, and 13.3 times higher than under R. When mildewed plants were exposed to cycles of 18 h of W light followed by 6 h of B, R, or F-R light, or darkness, R reduced the number of conidia trapped by ~88% compared with darkness or F-R. Interrupting the dark period with 1 h of R light reduced the number of conidia trapped, while 1 h of F-R following the 1 h of light from R nullified the suppressive effect of R.	[38]
*Rosa* ×*hybrida* L.	Control (no supplemental lighting); downward lighting at 150 μmol⋅m^−2^⋅s^−1^; and upward lighting at 150 μmol⋅m^−2^⋅s^−1^.	Control decreased flower number and lower-leaf senescence. Downward LED lighting promoted blooming and lower-leaf senescence. Upward LED lighting promoted blooming and maintained the photosynthetic abilities of the leaves, including the lower leaves.	[39]
*Salvia nemorosa* L. ‘Lyrical Blues’, *Gaura lindheimeri* Engelm. and Gray ‘Siskiyou Pink’	[R (660 nm)]:[B (460 nm)] light ratios (%) of 100:0 (R100:B0), 75:25 (R75:B25), 50:50 (R50:B50), or 0:100 (R0:B100).	All light-quality treatments did not change callus diameter and rooting percentage. R75:B25 or R50:B50 increased relative leaf chlorophyll content. R50:B50 decreased stem lengths of both species’ cuttings, and increased the root biomass compared to SL.	[40]
*Tagetes tenuifolia* Cav., *Celosia argentea* L.	RB: 65% R, 35% B; RGB: 47% R, 19% G, 34% B; and different photosynthetic photon flux densities (110, 220, and 340 µmol m^−2^ s^−1^).	Lowest level of photosynthetically active photon flux (110 µmol m^−2^ s^−1^) reduced growth and decreased the phenolic contents in all species. Total carotenoid content and antioxidant capacity were enhanced by the middle intensity (220 µmol m^−2^ s^−1^), regardless of spectral combination. The inclusion of green light at 340 µmol m^−2^ s^−1^ in the RB increased the growth (dry weight biomass) and the accumulation of bioactive phytochemicals.	[41]
*Tradescantia zebrina* Bosse *Chlorophytum comosum* (Thunb.) Jacques	Different light treatments: TO Tube luminescent Dunn (TLD) lamps or control, TB (TLD lamps + blue light-emitting diodes (LEDs)), TR (TLD lamps + red LEDs), and TBR (TLD lamps + blue and red LEDs).	Both species had increased root, shoot, and total dry weights under blue LED conditions. The chlorophyll concentration showed a specific response in each species under monochromic or mixed red–blue LEDs. The highest photosynthetic rate was measured under the addition of mixed red–blue LEDs with TLD lamps. The addition of blue LEDs increased the production of ornamental foliage species.	[42]

Abbreviation: B = blue; R-R = far-red; G = green; R = red; W = white LED color; D = darkness.

## Data Availability

Not applicable.

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
