# Peer review of "LED Lighting to Produce High-Quality Ornamental Plants"

_plants, 2023, doi:10.3390/plants12081667_

Round 1
Reviewer 1 Report
Comments from Reviewer
The review is about the use of LED light for the high production of quality ornamental plants. But there are issues with the manuscript (MS) that must be addressed.
Line 15. Please see “proving” is correct or not.
L 42. Insert “to” before “find”.
L 63. Replace “do” with “due”.
L 115. Change “also” into normal form.
L 87. The works referred to by the authors in the table are less. Kindly incorporate more works performed recently.
L 136-140. The authors can simplify English writing.
L 140. What is “Blu”. Change it if it is wrong.
L 150-152. Change the style of English writing.
L 152-153. The sentences carry no meaning. Kindly change it to get the correct sense.
* The authors are requested to incorporate in the MS a schematic diagram illustrating how light quality affects the flowering in ornamental plant
L 168. may be “attributed”.
L 189-192. The sentences look confusing. Kindly change it to make it more understandable.
L 199 – 201. Pls change the English style of writing. Don’t make things complicated by mixing forcefully different sentences.
L 216. Change “leaves” to “leaf”.
L 217-220. Pls put commas wherever necessary. The sentences cannot be written continuously without putting commas at appropriate positions.
L 232. Change “affects” to “affect”
L 239 . Change “ modulate” to “ modulated” and “influence” to “influenced” .
L 240. “affects” should be “affect”. Pls understand the correctness of the English grammar of the sentence.
L 246. What is “impatiens” ?
L 260. Change “ ensure” to “ensures”
L 270. Put comma after cutting. Thye author"s missed many commass in the other sentences as well.
L 275. Replace “strongly” with “ strong”.
L 284. “ tend” to “tends”
L 302. “ extreme” to “ extremely”
L 344. Put “to” after “ also”.
L 356. Put a reference after “ hostharvest”.
L 376. Change “leaves” to “leaf”
L 386-389. The sentences look very confusing. Simplify it.
L 391. Put a comma after “flowers”
L 392. Put a comma after “plants”
L 392-394. Change the English writng style, as there is no proper structuring of the sentences.
L 397. Put comma after “plants”
L 409. Put references after “dispersions”
L 414. Change “cause” to “causing”
L 416. Change “reduced” to “reduce”/ Authors should understand that the first form of a verb follows after the infinitive “to”
L 403 -438. There are many commas missing in sentences. Kindly put it in appropriate positions to make sentences readable and understandable. These mistakes are also made in other parts of the MS.
The authors are requested to look into all the issues raised by the reviewer in the MS. There are many grammatical errors in the MS which need to be fixed before the MS is approved for publication. The authors may use Grammarly or take help from an English language expert to address the above-mentioned issues.
Author Response
Reviewer 1
Dear reviewer,
The authors would like to thank you for your comments. The manuscript has been accordingly revised. Corrections and suggestions have been implemented in the current version of the manuscript. The manuscript has also undergone by MPDI English pre-edit services. All the modifications are highlighted in yellow in the manuscript. We hereby provide a point-by-point answer.
The authors
Comments from Reviewer
The review is about the use of LED light for the high production of quality ornamental plants. But there are issues with the manuscript (MS) that must be addressed.
Line 15. Please see “proving” is correct or not.
Author Answer (A.A.): sorry for the mistake; we inserted “providing” in the text
L 42. Insert “to” before “find”.
A.A.: Done
L 63. Replace “do” with “due”.
A.A.: Done
L 115. Change “also” into normal form.
A.A.: Done
L 87. The works referred to by the authors in the table are less. Kindly incorporate more works performed recently.
A.A.: Some new works are incorporated in the table
L 136-140. The authors can simplify English writing.
A.A.: As suggested, the sentence was modified. The manuscript has also undergone by MPDI English pre-edit services.
L 140. What is “Blu”. Change it if it is wrong.
A.A.: The correct word is blue and it has been corrected.
L 150-152. Change the style of English writing.
A.A.: As suggested by the referee, the sentence was modified.
L 152-153. The sentences carry no meaning. Kindly change it to get the correct sense.
A.A.: As suggested by the referee, the sentence was modified
* The authors are requested to incorporate in the MS a schematic diagram illustrating how light quality affects the flowering in ornamental plant
A.A.: a schematic diagram has added (Figure 2).
L 168. may be “attributed”.
A.A.: Done.
L 189-192. The sentences look confusing. Kindly change it to make it more understandable.
A.A.: As suggested by the reviewer, the sentence was modified.
L 199 – 201. Pls change the English style of writing. Don’t make things complicated by mixing forcefully different sentences.
A.A.: As suggested by the reviewer, the sentence was simplified.
L 216. Change “leaves” to “leaf”.
A.A.: Done.
L 217-220. Pls put commas wherever necessary. The sentences cannot be written continuously without putting commas at appropriate positions.
A.A.: As suggested by the referee, the commas were inserted.
L 232. Change “affects” to “affect”
A.A.: Done.
L 239. Change “modulate” to “modulated” and “influence” to “influenced”.
A.A.: Done.
L 240. “affects” should be “affect”. Pls understand the correctness of the English grammar of the sentence.
A.A.: As suggested by the referee, the sentence was rewritten.
L 246. What is “impatiens”?
A.A.: As suggested by the reviewer, the sentence was modified, adding the scientific name of the plant (Impatiens hybrid hort.)
L 260. Change “ensure” to “ensures”
A.A.: Done.
L 270. Put comma after cutting. The author’s missed many commas in the other sentences as well.
A.A.: As suggested by the reviewer, the commas were checked throughout the manuscript.
L 275. Replace “strongly” with “strong”.
A.A.: Done.
L 284. “tend” to “tends”
A.A.: Done.
L 302. “extreme” to “extremely”
A.A.: Done.
L 344. Put “to” after “also”.
A.A.: Done.
L 356. Put a reference after “hostharvest”.
A.A.: The reference “Aalifar, M.; Aliniaeifard, S.; Arab, M.; Zare Mehrjerdi, M.; Dianati Daylami, S.; Serek, M.; Woltering, E.; Li, T. Blue light improves vase life of carnation cut flowers through its effect on the antioxidant defense system. Front. Plant Sci. 2020, 11, 511. https://doi.org/10.3389/fpls.2020.00511” was added.
L 376. Change “leaves” to “leaf”
A.A.: Done.
L 386-389. The sentences look very confusing. Simplify it.
A.A.: Done.
L 391. Put a comma after “flowers”
A.A.: Done.
L 392. Put a comma after “plants”
A.A.: Done.
L 392-394. Change the English writing style, as there is no proper structuring of the sentences.
A.A.: Done.
L 397. Put comma after “plants”
A.A.: Done.
L 409. Put references after “dispersions”
A.A.: The reference “Erb, M.; Kliebenstein, D. J. Plant secondary metabolites as defenses, regulators, and primary metabolites: the blurred functional trichotomy. Plant Physiol. 2020, 184(1), 39-52” was added.
L 414. Change “cause” to “causing”
A.A.: Done.
L 416. Change “reduced” to “reduce”/ Authors should understand that the first form of a verb follows after the infinitive “to”
A.A.: Sorry for the mistake; the correction was done.
L 403 -438. There are many commas missing in sentences. Kindly put it in appropriate positions to make sentences readable and understandable. These mistakes are also made in other parts of the MS.
A.A.: As suggested by the reviewer, we added commas throughout the manuscript.
The authors are requested to look into all the issues raised by the reviewer in the MS. There are many grammatical errors in the MS which need to be fixed before the MS is approved for publication. The authors may use Grammarly or take help from an English language expert to address the above-mentioned issues.
A.A.: as already mentioned, the manuscript has also undergone by MPDI English pre-edit services.

Reviewer 2 Report
The MS "LED lighting to produce high-quality ornamental plants' is an attempt to summarize potential LED benefits to floriculture industry. The paper is fine while has some shortcomings.
1. English should be checked, for example see line 62 - due to (not do to), line 299 - modulated/influenced, 240 - affects, 299 - such as, 302 - extremely, 306 - product type, 316 - accelerates.
2. Figure 1 legend (B, C, D) does not correspond to the figure.
3. Long parts of the text (lines 93-132, 313-336) are not actually related to LED lighting and seem to be excessive.
4. Line 134 and throughout the text: as SD and LD plants abbreviations introduced, use them.
5. Lines 241-242 - compared to what?
6. Line 241 - How the control of transpiration is related to the Plant architecture (subtitle)? Lines 258-260 are also not about plant architecture.
7. Lines 166-167 - affected by what?
Author Response
Reviewer2
Dear reviewer,
The authors would like to thank you for your comments. The manuscript has been accordingly revised. Corrections and suggestions have been implemented in the current version of the manuscript. All the modifications are highlighted in yellow in the manuscript. We hereby provide a point-by-point answer.
The authors
The MS "LED lighting to produce high-quality ornamental plants' is an attempt to summarize potential LED benefits to floriculture industry. The paper is fine while has some shortcomings.
- English should be checked, for example see line 62 - due to (not do to), line 299 - modulated/influenced, 240 - affects, 299 - such as, 302 - extremely, 306 - product type, 316 - accelerates.
Author Answer (A.A.):
- Figure 1 legend (B, C, D) does not correspond to the figure.
A.A.: According to the reviewer comment, the figure legend was corrected.
- Long parts of the text (lines 93-132, 313-336) are not actually related to LED lighting and seem to be excessive.
A.A.: We tried our best to synthetize it.
- Line 134 and throughout the text: as SD and LD plants abbreviations introduced, use them.
A.A.: As suggested by the reviewer the abbreviations were added.
- Lines 241-242 - compared to what?
A.A.: As suggested by the reviewer, we modified the sentence.
- Line 241 - How the control of transpiration is related to the Plant architecture (subtitle)?
A.A.: We thought that worth to mention in plant architecture, the susceptibility of cuttings to fast drying as a background notion linked to their overall quality, since cuttings is a widely spread procedure and a well formulated light recipe might be useful to control the transpiration process.
Lines 258-260 are also not about plant architecture.
A.A.: As suggested by the reviewer, we deleted this sentence.
- Lines 166-167 - affected by what?
A.A.: As suggested by the reviewer, we specified end-of-day far-red treatment in the sentence.

Reviewer 3 Report
In my opinion the manuscript is interesting, well written and organized. Some minor spell checks are required.
Author Response
Reviewer 3
Dear reviewer,
The authors would like to thank you for your comments. The manuscript has been revised. Corrections and suggestions have been implemented in the current version of the manuscript. All the modifications are highlighted in yellow in the manuscript.
The authors
In my opinion the manuscript is interesting, well written and organized. Some minor spell checks are required.
A.A.: very thanks for your opinion; the manuscript has undergone by MPDI English pre-edit services.

Round 2
Reviewer 1 Report
The authors have addressed the issues raised by the Reviewer and appropriately modified in the Manuscript. The manuscript is more refined and can be accepted for publication.